# Generation of Correction Data for Autonomous Driving by Means of Machine Learning and On-Board Diagnostics

**DOI:** 10.3390/s23010159

**Published:** 2022-12-23

**Authors:** Alberto Flores Fernández, Eduardo Sánchez Morales, Michael Botsch, Christian Facchi, Andrés García Higuera

**Affiliations:** 1Technische Hochschule Ingolstadt, Esplanade 10, 85049 Ingolstadt, Germany; 2Escuela Técnica Superior de Ingeniería Industrial, Universidad de Castilla-La Mancha, Calle Altagracia 50, 13001 Ciudad Real, Spain; 3European Parliamentary Research Service, Rue Wiertz 60, B-1047 Brussels, Belgium

**Keywords:** On-Board Diagnostics, Machine Learning, Transformer Neural Network, Autonomous Driving, ADAS, Inertial Navigation Systems

## Abstract

A highly accurate reference vehicle state is a requisite for the evaluation and validation of Autonomous Driving (AD) and Advanced Driver Assistance Systems (ADASs). This highly accurate vehicle state is usually obtained by means of Inertial Navigation Systems (INSs) that obtain position, velocity, and Course Over Ground (COG) correction data from Satellite Navigation (SatNav). However, SatNav is not always available, as is the case of roofed places, such as parking structures, tunnels, or urban canyons. This leads to a degradation over time of the estimated vehicle state. In the present paper, a methodology is proposed that consists on the use of a Machine Learning (ML)-method (Transformer Neural Network—TNN) with the objective of generating highly accurate velocity correction data from On-Board Diagnostics (OBD) data. The TNN obtains OBD data as input and measurements from state-of-the-art reference sensors as a learning target. The results show that the TNN is able to infer the velocity over ground with a Mean Absolute Error (MAE) of $0.167\;\frac{km}{h}$ ($0.046\;\frac{m}{s}$) when a database of 3,428,099 OBD measurements is considered. The accuracy decreases to $0.863\;\frac{km}{h}$ ($0.24\;\frac{m}{s}$) when only 5000 OBD measurements are used. Given that the obtained accuracy closely resembles that of state-of-the-art reference sensors, it allows INSs to be provided with accurate velocity correction data. An inference time of less than 40 ms for the generation of new correction data is achieved, which suggests the possibility of online implementation. This supports a highly accurate estimation of the vehicle state for the evaluation and validation of AD and ADAS, even in SatNav-deprived environments.

## 1. Introduction

AD and ADAS are popular trends in the automotive industry. Both AD and ADAS can serve to provide driving comfort and fuel efficiency for long-haul trips, motion prediction [1], or occupant protection functions. Whichever the intended use may be, vehicles can cause serious harm to humans in the case of a malfunction. This leads to the fact that all vehicle functions that have an effect on the longitudinal or lateral vehicle dynamics have to be extensively tested. The required AD and ADAS testing in turn demands a highly accurate vehicle state that serves as a reference for an objective evaluation and validation.

A widely accepted method to generate such a highly accurate vehicle state is the use of INSs that combine the measurements from inertial sensors with correction data from SatNav receivers. An advantage of this method is that even consumer-grade SatNav receivers without correction data are able to deliver acceptable velocity and COG correction data. Furthermore, the United States Department of Defense supports the continuous development of the GPS III system by assigning contracts to Space Exploration Technologies Corp and Lockheed Martin [2]. This means that new generation SatNav receivers should be able to improve their positioning accuracy from the current 2–5 m to around 30 cm, which should enable accurate lane navigation [3].

Even with all the Research and Development (R&D) in the field of SatNav, there are still situations that represent challenges. For example, consumer-grade SatNav receivers usually have a 1 Hz update rate, which might not be fast enough for certain automotive research applications, as could be emergency braking or emergency lane change maneuvers. A challenge that both consumer-grade and high-end SatNav receivers face is the multipath effect. This appears when the SatNav signals do not follow a straight path from the satellites to the receiver but bounce off other objects before. This can happen while driving in urban canyons, where the SatNav signals bounce off the windows of tall buildings. Another situation where the multipath effect can arise is while driving next to trailers, where the metallic containers are prone to reflect the SatNav signals as well. Most SatNav receivers output a navigation solution that can include the position, velocity over ground, and COG of the receiver, as well as a degradation metric for the navigation solution, as are the dilution of precision or the standard deviation. The multipath effect alters the navigation solution, causing the measured state variables to diverge from the true vehicle state. The multipath effect further complicates the SatNav by also altering the degradation metric of the navigation solution. Furthermore, given that this metric is precisely what many sensor fusion techniques rely on, the multipath effect negatively affects the estimated vehicle state as well. Even though there are many approaches found in the literature that address the multipath effect [4,5,6,7,8], these usually require to read the pseudo-ranges of the SatNav signals. On the contrary, most SatNav receivers do not output pseudo-ranges, but only the navigation solution.

Another challenge that all SatNav receivers face is the absence of SatNav coverage. This problem occurs mostly while driving in closed-sky areas, such as tunnels and multi-story parking structures, but can arise as well while driving on places where the sky is not completely obstructed, such as tree-flanked roads. The absence of SatNav coverage forces other fallback solutions to be adopted, such as the Dead-reckoning or the replacement of the SatNav receiver with other external sensors. The issue of SatNav outage is augmented in the cases where the outage is intermittent, mainly because the alternating availability of SatNav reception usually comes with a wrongful estimation of the measured state variables, which in turn can cause oscillations, jumps, and instability in general with the navigation solution. Similarly to what happens when the multipath effect is present, the degradation metric of the navigation solution can be wrongly estimated, which again ends up in an incorrect estimation of the vehicle state.

Given that road vehicles are equipped with a wide variety of sensors that constantly monitor the vehicle state, they are interesting candidates for the generation of correction data for INSs. As their sensors are mounted on the vehicle itself, such as hall sensors on the wheels, the state variables of the vehicle can be estimated without requiring external information. Furthermore, even though many research works include test vehicles, most of them are specially prepared platforms that either have equipped high-end external sensors, or are built with sensors whose measuring performance is much better than those present in commercial vehicles. The present research work addresses precisely this point: the generation of correction data for INSs from OBD data from commercial vehicles for a highly accurate and robust vehicle state estimation that can serve as a reference for the evaluation and validation of AD and ADAS.

With regards to the velocity, it should be noted that the cost per part plays an important role when vehicle manufacturers choose the sensors to install in their vehicles. Even though a couple of cents per sensor might not sound representative on a first instance, once one considers the economy of scale and the number of vehicles produced, it does make sense to consider the sensor price especially when a kilometer-per-hour-accurate sensor perfectly fulfills consumer expectations and the safety requirements. This leads to the fact that not every vehicle on the road has exactly the same sensors installed, which means that the accuracy of the velocity information obtained from the OBD can vary from vehicle to vehicle. Even when the error of the OBD velocity could be modeled for a specific vehicle, it would not be correct to apply this same model to the rest of vehicles. It is also known that the calibration of the velocity sensors of a vehicle can change over its lifetime, thus implying different error models for the same vehicle.

Despite their disadvantages, the OBD data are a valuable source of information because of three key aspects: (1) availability, (2) accessibility, and (3) resources. Regarding the availability, a great majority of vehicles on the road are compliant with the OBD-II protocol, which standardizes (among other things) its hardware interface and the information transmission. This means that the hardware and algorithms can be developed generically for all vehicles that comply with the OBD-II standard, instead of vehicle-specific solutions. As for the accessibility, the OBD-II protocol defines the access to certain vehicle information. Consequently, the information access is not vehicle-specific either, but also possible for all vehicles compliant with the OBD-II standard. Finally, both the financial and human resources play an important role as well. Even though there are high-end sensors in the market that provide highly accurate velocity information, these imply an important financial cost. Additionally, as the number of used sensors increases, so does the testing complexity. This includes the time required for the test preparation as well as the time needed to evaluate the recorded data. The reasons detailed above strongly support the use of ML-methods to generate correction data for INSs out of the OBD data, as this addresses the main challenge: the ability to adapt.

To the best of the authors’ knowledge there is no previous investigation that makes use of TNNs, OBD data as input to infer velocity over ground data, and that uses state-of-the-art reference sensors for the training and evaluation of the TNN.

This research work makes the following contributions: (1) proposes an ML-method to generate highly accurate velocity correction data from real-world OBD data and (2) performs various ablation studies that confirm the robustness of the method against variations on the database size and of the input sequence length.

The rest of the paper is structured as follows. In Section 2, related works regarding the use of ML for automotive applications are presented. In Section 3, the research framework, the dataset generation and processing, and the used ML-model are described. In Section 4, an evaluation of the results of the proposed methods is shown. In Section 5, a discussion of the results is presented. The paper is concluded in Section 6.

## 2. Related Works

As stated above, AD and ADAS have a wide variety of applications. Furthermore, with the ever growing popularity of ML, the application of ML-methods in the automotive field has increased as well over the past years. In [9], the authors propose a classifier-based method for the segmentation of AD maneuvers (parallel- and cross-parking). The authors generate a training dataset by means of computer simulations and validate their proposed method with a small-scale autonomous vehicle. This demonstrates that ML-algorithms can be implemented in resource-limited platforms. Another example of research on vehicle trajectory planning, but for safety critical applications, is shown in [10]. There, the authors reduce the runtime of trajectory-planning algorithms by replacing certain computationally intensive modules of the trajectory planners with ML- and analytical methods. The optimized trajectory planing algorithms are implemented on various hardware platforms so as to compare the runtime. Similar as in [9], the authors of [10] demonstrate that ML-algorithms can run in real-time, even in resource-limited microcontrollers. Another research work that uses ML-methods for the estimation of the vehicle state is presented in [11]. There, the authors train with real data a Random Forest to detect when a vehicle is standing still. The relevance of this method is that it uses only inertial measurements, which allow the method to function without external sensors, such as the SatNav or the OBD data. The standstill detection is relevant because it helps to avoid the divergence of the estimated vehicle state when the vehicle is not moving. Without such a classifier, the continuous integration of the inertial measurements leads to an apparent ever-increasing velocity over ground, which in turn distorts the rest of the estimated state vector. The methodology of [11] also deals with the fusion of the inertial signals with vehicle motion models in order to improve the accuracy of the vehicle state during Dead-reckoning. More methods that combine mathematical modeling with ML-methods can be found in [12].

One key element that is required from both AD and ADAS in order to plan safe trajectories is to know where the road is (a) drivable and (b) available. The first refers to the road infrastructure: lanes and sidewalks, for example. The availability refers to the absence of obstacles on the drivable road. The authors of [13] deal with both situations. They propose an ML-method that predicts the availability of the road in the next few seconds of a given traffic scenario. The input of their algorithm is the current state of a given traffic scenario and the output is a probabilistic space-time representation of the traffic scenario termed Predicted Occupancy Grids (POGs). The POGs represent the probability that certain road grids will be occupied in the next few seconds.

Another examples of the use of ML-methods for safety-critical applications are the products of MobilEye. These are cameras that can be retro-fitted on the windshield of road vehicles. The cameras face in the driving direction and are able to detect vehicles and to output a “Time to collision” that is coupled with acoustic signals to prevent the drivers to take action. More information about these devices can be found on [14].

Even in in-city situations, road vehicles are allowed to reach velocities of up to $80\;\frac{km}{h}$ in certain situations [15]. This, combined with the close proximity between vehicles, can lead to unavoidable traffic accidents. A common practice to estimate the severity of a vehicle crash by non-destructive means is the use of Finite Element Method (FEM). These simulations provide a highly accurate depiction of a vehicle crash, including the vehicle deformation, but their computational cost implies that a single FEM simulation can take several hours. The authors of [16] propose a Crash Severity Predictor that is trained with FEM crash simulation data to predict the crash severity distribution of an imminent collision. The prediction has an accuracy between 85% and 98% and is performed in ≈0.2 s, which is many orders of magnitude faster than an FEM simulation. An accurate estimation of the crash severity helps to decide whether the vehicle occupants should be prepared for an imminent collision (for example, with seat belt pretensioners) and aids as well to decide if non-reversible safety features should be triggered, as are the airbags.

The research performed in [17] marked a turning point in the field of Natural Language Processing (NLP) by proposing the “Attention” mechanism and the ML-architecture named “Transformers”. This work was extended to other fields of application, such as time series or sequence-to-sequence prediction. In this sense, works such as [18,19] showed the potential of TNNs in the field of predicting the motion of traffic participants.

The research work that most resembles the methods proposed herein is shown in [20]. There, the authors train an Adaptive Neuro-Fuzzy Inference System (ANFIS) that receives six inertial measurements as an input—the angular velocities around and the accelerations along the *x*, *y*, and *z* axes of a consumer-grade Inertial Measurement Unit (IMU)—and the learning targets are the same inertial measurements but from a high-end IMU. The authors evaluate the performance of their method by comparing the navigation solution as estimated with three sources of information: the measurements from the consumer-grade IMU without ANFIS, the measurements from the consumer-grade IMU with ANFIS, and the measurements from the high-end IMU. Their results show an improvement of 70% in the 2D positioning and 92% improvement in the velocity estimation, which shows that consumer-grade sensors can approach the measuring characteristics of their high-performance counterparts. The main differences between [20] and the present work are (1) the sensors used, (2) the ML-model applied, (3) the state variable to address, and (4) the analysis of the possibility for online implementation. With regards to (1), the present work does not use external sensors (as is the consumer-grade IMU), but only the on-board sensors of commercial vehicles instead. The difference is relevant because, contrary to external sensors, the OBD sensors behave like a black box, where the sensor specifications, including the update rate, sensor bias, and sensor noise are not known, which complicates, among other things, the parametrization of sensor fusion algorithms and sensor filters. With regards to (2), the present work makes use of TNN as the ML-model for the generation of correction data from OBD data. As reported in [21], the use of ANFIS implies a high training complexity, is computationally expensive, and poses a challenge for applications with large datasets. The present work additionally provides several ablation studies using TNNs. With respect to (3), the present work deals with the generation of velocity correction data, instead of accelerations and angular velocities. The difference is relevant because, as shown in [11], by directly generating the velocity correction data, one is able to improve the accuracy of the navigation solution when compared to the one obtained from the integration of inertial measurements. Finally, with regards to (4), the present work analyses as well the possibility of implementing the proposed methods for online processing. This increases the relevance of the proposed methods, as it better supports future research works than methods that only work offline. This also provides information about the possibility of implementing the proposed methods in embedded hardware for real-time applications.

## 3. Materials and Methods

This section is structured as follows. First, the mathematical preamble and description of the reference sensors are given in Section 3.1. The data generation and processing are detailed in Section 3.2 and Section 3.3, respectively. Afterwards, the ML-model based on TNN, for the generation of correction data, is presented in Section 3.4.

### 3.1. Research Framework

The land vehicles move on the Local Tangent Plane (LTP). This is a Cartesian reference frame composed of the mutually perpendicular $x_{\mathrm{LTP}}$, $y_{\mathrm{LTP}}$ and $z_{\mathrm{LTP}}$ axes, with $\vec{z_{\mathrm{LTP}}}\;=\;\vec{x_{\mathrm{LTP}}}\;\times \;\vec{y_{\mathrm{LTP}}}$, and with origin $o_{\mathrm{LTP}}$ at an arbitrary location on the surface of the Earth. The axis orientation of the LTP is similar to the East–North–Up reference frame, where the $x_{\mathrm{LTP}}\;\times \;y_{\mathrm{LTP}}$ plane is perpendicular to the gravitational pull of the Earth, the $y_{\mathrm{LTP}}$ axis is pointing to the true north of the Earth, and the $z_{\mathrm{LTP}}$ axis is parallel to the gravitational pull of the Earth but positive upwards.

The Local Car Plane (LCP) is the vehicle reference frame and is defined analogue to the ISO 8855:2011 norm [22]. The LCP is composed of the mutually perpendicular $x_{\mathrm{LCP}}$, $y_{\mathrm{LCP}}$, and $z_{\mathrm{LCP}}$ axes, with $\vec{z_{\mathrm{LCP}}}\;=\;\vec{x_{\mathrm{LCP}}}\;\times \;\vec{y_{\mathrm{LCP}}}$, and origin $o_{\mathrm{LCP}}$ at the Center of sprung Mass (CoM) of the vehicle. The $x_{\mathrm{LCP}}$ axis is parallel to the longitudinal axis of the vehicle and points towards the hood, the $y_{\mathrm{LCP}}$ axis is parallel to the transversal axis of the vehicle and points towards the driver seat, and the $z_{\mathrm{LCP}}$ axis points upwards. This coordinate system is illustrated in Figure 1.

All quantities are expressed in SI units (*Système international d’unités*) unless otherwise specified.

The state vector $x_{s}$ of a land vehicle can be defined as follows
(1)$$\begin{array}{c} x_{s}={\left[\begin{array}{cccccccc} x_{s} & y_{s} & \theta_{z,s} & {\dot{\theta }}_{z,s} & v_{s} & \beta_{s} & a_{x,s} & a_{y,s} \end{array}\right]}^{T}, \end{array}$$
where $\left(x_{s},y_{s}\right)$ are the (x,y) coordinates of the vehicle in LTP, $\theta_{z,s}$ is the vehicle orientation in LTP, ${\dot{\theta }}_{z,s}$ is the angular velocity of the vehicle around the $z_{\mathrm{LCP}}$-axis (yaw rate), $v_{s}$ is the velocity over ground of the vehicle, $\beta_{s}$ is the side-slip angle of the vehicle in LCP, $a_{x,s}$ is the vehicle acceleration along the $x_{\mathrm{LCP}}$-axis, and $a_{y,s}$ is the vehicle acceleration along the $y_{\mathrm{LCP}}$-axis.

In order to perform an objective and accurate evaluation of the OBD-obtained data and of the inference of the TNN, two reference sensors are used: the Automotive Dynamic Motion Analyzer (ADMA) and the Correvit S-Motion. The ADMA is an INS that is equipped with servo-accelerometers and optical gyroscopes, and can receive SatNav correction data such as what is known as “Real-Time-Kinematic (RTK)” [23]. RTK refers to when the SatNav receiver not only obtains information from the satellites, but from a radio antenna as well. The ADMA-RTK system can output a centimeter-accurate position, a velocity over ground with an accuracy of $0.03\;\frac{km}{h}$, and COG with an accuracy of 0.005${}^{\circ }$ [24]. The ADMA-RTK system can currently be used for the ABS/ESP ISO 26262 certification. In addition, GeneSys Elektronik, the manufacturer, was certified in 2022 to perform calibrations according to the ISO-17025 [25] standard and is thus a calibration laboratory accredited by DAkkS. This means that ADMA systems are internationally traceable and can be used as reference sensors by expert organizations such as TÜV, BAST, or DEKRA, for example for acceptance tests, which further confirms the suitability of the ADMA as a reference sensor.

The ADMA is capable of generating a measurement vector $z_{r}$ so that
(2)$$\begin{array}{c} z_{r}={\left[\begin{array}{cccccccc} x_{r} & y_{r} & \theta_{z,r} & {\dot{\theta }}_{z,r} & v_{r} & \beta_{r} & a_{x,r} & a_{y,r} \end{array}\right]}^{T}, \end{array}$$
where $\left(x_{r},y_{r}\right)$ are the (x,y) coordinates of the vehicle in LTP, $\theta_{z,r}$ is the vehicle orientation in LTP, ${\dot{\theta }}_{z,r}$ is the angular velocity of the vehicle around the $z_{\mathrm{LCP}}$-axis (yaw rate), $v_{r}$ is the velocity of the vehicle over the ground, $\beta_{r}$ is the side-slip angle of the vehicle, $a_{x,r}$ is the vehicle acceleration along the $x_{\mathrm{LCP}}$-axis, and $a_{y,r}$ is the vehicle acceleration along the $y_{\mathrm{LCP}}$-axis.

The ADMA generates the measurement vector $z_{r}$ by using statistical filtering methods to combines the measurements from the own inertial sensors (${\dot{\theta }}_{z,r}$, $a_{x,r}$ and $a_{y,r}$) with measurements from external sensors. Due to its high accuracy, practicality, and accessible cost for the end-user, the preferred external sensors are the SatNav receivers. These deliver measurements of the own position ($x_{G}$, $y_{G}$), COG, and velocity over ground ($v_{G}$). It should be noted that when $\beta_{s}\to 0,\theta_{z,r}\to COG$, which means that under traction driving [26], the COG as measured by the SatNav can be used as correction data $\theta_{z,r}$ for $\theta_{z,s}$.

One disadvantage of using SatNav is that there are situations where it is not available, such as tunnels, parking structures, and urban canyons, among others. This means that even with the state-of-the-art inertial sensors, such as the ones installed in the ADMA, a divergence in the state vector that grows over time is unavoidable if no correction data are available.

An alternative to the use of SatNav as correction data is the Correvit S-Motion, which is a state-of-the-art sensor that consists of a camera mounted on the bodywork of the vehicle that points downwards. It estimates the linear velocities along its *x* and *y* axes by means of an optical grid method. This method consists of detecting the motion over time of certain patterns that are captured with the camera. Like GeneSys Elektronik, the manufacturer of the Correvit S-Motion, Kistler, was accredited in 2022 by DAkkS as a calibration laboratory [27], which confirms the adequacy of the Correvit S-Motion to serve as a reference sensor. Further details of this method can be found in [28]. For simplification purposes, it is assumed that the *x* and *y* axes of the Correvit are parallel to the $x_{\mathrm{LCP}}$ and $y_{\mathrm{LCP}}$ axes, respectively. The Correvit is able to measure the linear velocities $v_{x,\mathrm{LCP}}$ and $v_{y,\mathrm{LCP}}$ along the $x_{\mathrm{LCP}}$ and $y_{\mathrm{LCP}}$ axes with an accuracy of $\pm 0.2\;\frac{km}{h}$ [29]. With $v_{x,\mathrm{LCP}}$ and $v_{y,\mathrm{LCP}}$, the velocity of the vehicle over ground $v_{r}$ can be calculated as follows
(3)$$\begin{array}{c} v_{r}=\sqrt{v_{x,\mathrm{LCP}}^{2}+v_{y,\mathrm{LCP}}^{2}}. \end{array}$$

In order to guarantee the robustness of the reference data, the Correvit delivers the velocity correction data to the ADMA, which fuses the measurements of both. The velocity $v_{r}$ that results from the fusion of the ADMA, SatNav, and Correvit data serves as the reference used in this research work.

### 3.2. Data Generation

In order to collect both the OBD and the reference data, a test vehicle (shown in Figure 2) is equipped with (1) state-of-the-art reference sensors (the ADMA and the Correvit), and (2) a car PC with PCI CAN interfaces that records both the OBD and the reference data. Given that the car PC is equipped with PCI interfaces, one is able to avoid the unknown and random data transmission delays that are typical of the USB protocol.

The algorithms needed to generate the dataset are implemented in Robotic Operating System (ROS), which runs on Linux. The process consists of (1) acquiring the raw data from both the OBD and reference sensors, (2) parsing the raw data of both the OBD and the reference sensors, (3) storing of both raw and parsed data in rosbags, (4) exporting rosbags to Comma-Separated Values (CSV) files, and (5) data preparation. The process is shown in Figure 3.

In order to acquire the raw OBD data, a custom cable is built that has an OBD-II plug on one side and a CAN sub-D9 plug on the other end. The cable is connected accordingly to the OBD-II port of the test vehicle and the car PC. Two ROS nodes are then implemented: one that reads the raw OBD data and one that parses them. There are two options to parse the OBD data: (A) by means of the OBD protocol and (B) by means of an open gateway. In the case of the OBD protocol, the J1979 standard [30] specifies the OBD-II PIDs (OBD-II Parameter IDentifications), which allows it to request certain diagnostic information from OBD-II compliant vehicles. The J1979 standard also defines how to parse the OBD data. The open gateway option applies to vehicles that are specially prepared for R&D, as can be prototypes, custom test vehicles, or mock-ups. In these cases, the OBD data can be parsed, for example, with a CAN matrix provided by the manufacturer of the test vehicle. Among others, the OBD measurements can include the velocity $v_{o}$ as shown by the speedometer, the individual velocity of each tire, and the steering angle. However, for the purpose of this paper, only $v_{o}$ is considered.

As for the reference sensor data, the Correvit is connected to the ADMA via a CAN cable and the ADMA is connected to the car PC via an Ethernet cable. The ADMA receives position, velocity, and COG correction data from a SatNav receiver, and velocity correction data from the Correvit as well. The ADMA then sends to the car PC the performed inertial measurements, the correction data as received by the external sensors (SatNav and Correvit), and the result of the fusion of the inertial measurements with the correction data. A ROS node that runs on the car PC then reads and parses the ADMA data, as shown in [31]. Once both the OBD and reference data are parsed, they are stored in form of rosbags. The rosbags are finally converted to CSV files according to [32].

The sensor synchronization is a key aspect of objective data evaluation because it allows one to make a temporal correlation of the data. Some external sensors, such as the ADMA and the Correvit, have the option to be connected to SatNav receivers that deliver the UTC-Time so that the performed measurements are timestamped. This allows one to have a common and universal time axis to compare the measurements from different sensors. The OBD data, on the other hand, do not have a timestamp as read from the OBD gateway, nor can they be connected directly to a SatNav receiver to synchronize the data, but they have to be timestamped by the external hardware that reads the OBD data. For this research work, the synchronization of all sources of information (OBD, ADMA, and Correvit) is carried out by means of the UNIX time. As stated above, the data of all sources of information are read and recorded by means of a ROS implementation, which allows one to timestamp the read information with the internal clock of the car PC. As long as the car PC has internet access, it synchronizes its internal clock with the UNIX time, thus providing an adequate alternative for the UTC-time. One important aspect to consider is that, even when PCI interfaces are used, given that the algorithms run on ROS and Linux, there are still operating system-related data transmission delays that are neither known nor controllable. It is only when one compares the timestamps of the recorded data (Unix from the car PC and UTC from the SatNav receivers) that a temporal misalignment of the data can be spotted. Whenever a synchronization is needed, the OBD measurements are associated with the reference measurements that are closest in time. The synchronization error $\eta_{t}$ that results is defined as the time difference between the synchronized OBD $v_{o}$ and the corresponding reference sensor $v_{r}$ measurements, so
(4)$$\begin{array}{c} \eta_{t}=\left|t_{i,\mathrm{OBD}}-t_{j,\mathrm{REF}}\right|. \end{array}$$

Due to the limitations of the CAN-Bus standard, there are time differences between the OBD-obtained measurements and those made by the reference sensors (ADMA and Correvit). This happens mainly because several electronic control units are connected to the same Bus and send information at the same time. Each CAN information package has an arbitration ID that is assigned according to the importance of the transmitted message. This arbitration ID determines which messages are transmitted over the Bus at any given time, while the rest of the messages are lost. If more important information than the vehicle velocity is transmitted over the Bus (as can be the braking information), the velocity information is lost, thus causing that the velocity information to not be read from the OBD at deterministic time intervals. Moreover, the number of sensors installed in the vehicles also varies, thus causing variations in the load of the CAN-Bus, with the corresponding uncertainties in regards to the latency of the OBD data.

To the best of our knowledge, there are no publicly available OBD datasets that (1) have enough data for the training of ML-algorithms and (2) that have highly accurate sensors as reference. Therefore, an in-house dataset of real sensor data is generated. Given that it has not yet been published and in order to offer a better understanding of the proposed method, a detailed description of the generated dataset and of its pre-processing is given in what follows.

For this, the test vehicle is randomly driven on the closed test track of the University of Applied Sciences of Ingolstadt (Technische Hochschule Ingolstadt), Germany, covering velocities from $0\;\frac{km}{h}$ to $50\;\frac{km}{h}$. The dataset has a duration of ≈1282 min, which accounts for ≈535 km, 7,691,147 reference measurements and 3,840,995 OBD measurements. So as foster the diversity of the dataset, the vehicle was driven without aids that help maintain a constant velocity, such as cruise control. In addition, random acceleration and braking instances were included. The reason for the different number of OBD and reference measurements is that the OBD data cannot be obtained in deterministic time intervals. Instead, the arbitration IDs and the CAN implementation determine when the OBD data are published. A histogram of the dataset is shown in Figure 4.

### 3.3. Data Processing

After the sensor dataset is generated, the data processing is performed. The objective of this is to generate an ML- database from the sensor dataset. The ML-database is used later to train and evaluate the TNN (Section 3.4). The process consists of three steps: (1) data selection (Section 3.3.1), (2) data balancing (Section 3.3.2), and (3) data splitting (Section 3.3.3). A graphic depiction of the process is shown in Figure 5 and is detailed in the following subsections.

#### 3.3.1. Data Selection

This step aims to generate a primary set of samples from the sensor dataset generated in Section 3.2. This groups OBD data with the corresponding reference sensor measurements and filter out-of-range samples. A graphic depiction of this is shown in Figure 6.

##### Data Loading

The first step is to load the sensor dataset $D_{\mathrm{raw}}$. This is composed by all the performed driving tests so that
(5)$$\begin{array}{c} D_{\mathrm{raw}}=D_{1}⋃D_{2}⋃\cdots ⋃D_{i}⋃\cdots ⋃D_{I}, \end{array}$$
where $D_{i}$ indicates the *i*-th driving test and *I* the total number of driving tests.

Each *i*-th driving test $D_{i}$ contains two strings: one for the OBD data ($T_{i,\mathrm{OBD}}$) and another for the reference sensor measurements ($T_{i,\mathrm{REF}}$), so that
(6)$$\begin{array}{c} D_{i}=\left\{T_{i,\mathrm{OBD}},T_{i,\mathrm{REF}}\right\}, \end{array}$$
where OBD denotes OBD data and REF indicates reference sensor measurements.

Each string *T* is a temporal ordered sequence of points p=(v,t), where *v* is a velocity measurement with its corresponding timestamp *t*. Each string $T_{\mathrm{OBD}}$ and $T_{\mathrm{REF}}$ is then defined as follows
(7)$$\begin{array}{c} T_{\mathrm{OBD}}=\left[p_{1,\mathrm{OBD}},p_{2,\mathrm{OBD}},\cdots ,p_{i,\mathrm{OBD}},\cdots ,p_{N_{\mathrm{OBD}},\mathrm{OBD}}\right], \end{array}$$
(8)$$\begin{array}{c} p_{i,\mathrm{OBD}}=\left(v_{i,\mathrm{OBD}},t_{i,\mathrm{OBD}}\right), \end{array}$$
(9)$$\begin{array}{c} T_{\mathrm{REF}}=\left[p_{1,\mathrm{REF}},p_{2,\mathrm{REF}},\cdots ,p_{j,\mathrm{REF}},\cdots ,p_{N_{\mathrm{REF}},\mathrm{REF}}\right],\;\mathrm{and} \end{array}$$
(10)$$\begin{array}{c} p_{j,\mathrm{REF}}=\left(v_{j,\mathrm{REF}},t_{j,\mathrm{REF}}\right), \end{array}$$
where $N_{\mathrm{OBD}}$ is the total number of OBD data points and $N_{\mathrm{REF}}$ is the total number of reference measurements.

##### Sample Generation

The next step is to generate samples from $D_{\mathrm{raw}}$ in order to train the ML-model. Each sample S=(χ,γ) is composed of an input sequence χ and a target sequence γ so that
(11)$$\begin{array}{c} \chi =[v_{1,\mathrm{OBD}},v_{2,\mathrm{OBD}},\ldots ,v_{i,\mathrm{OBD}},\ldots ,v_{L_{\mathrm{in}},\mathrm{OBD}}],\;\mathrm{and} \end{array}$$
(12)$$\begin{array}{c} \gamma =[v_{1,\mathrm{REF}},v_{2,\mathrm{REF}},\ldots ,v_{j,\mathrm{REF}},\ldots ,v_{L_{\mathrm{out}},\mathrm{REF}}], \end{array}$$
where $L_{\mathrm{in}}$ indicates the length of the input sequence and $L_{\mathrm{out}}$ the length of the target sequence.

Given that the objective is to use OBD data to infer future velocity measurements as if they were made by the reference sensors, the timestamp associated with the first velocity measurement of the target sequence must be bigger than the timestamp associated to the last element of the input sequence, so that
(13)$$\begin{array}{c} t_{i,\mathrm{OBD}}<t_{i+1,\mathrm{OBD}}<\ldots <t_{i-1+L_{\mathrm{in}},\mathrm{OBD}}<t_{j,\mathrm{REF}}<t_{j+1,\mathrm{REF}}<\ldots <t_{j-1+L_{\mathrm{out}},\mathrm{REF}}, \end{array}$$
where the subindex *i* indicates the *i*-th element of $T_{\mathrm{OBD}}$ and *j* indicates the *j*-th element of $T_{\mathrm{REF}}$.

In order to distribute the elements of $T_{\mathrm{OBD}}$ and $T_{\mathrm{REF}}$ among all samples *S*, first, $L_{\mathrm{in}}$ is set to an arbitrary value. $T_{\mathrm{OBD}}$ is then split into $N_{\mathrm{seq}}=\left\lfloor \frac{N_{\mathrm{OBD}}}{L_{\mathrm{in}}}\right\rfloor$ input sequences, where ⌊·⌋ indicates the floor operation and each input sequence is assigned to a sample. The target sequence of the *i*-th sample is composed of $L_{\mathrm{out}}$ elements of $T_{\mathrm{REF}}$ with a timestamp between that of the last element of the *k*-th and that of the first element of the k+1-th input sequences. A graphical depiction of the sample generation is shown in Figure 7.

##### Samples Filtering

In order to have a homogeneous and meaningful sample dataset, the previously generated samples are filtered according to the following criteria:Consistent length. Due to the fact that the TNN expects input and target sequences of fixed length, and to avoid sequence completion techniques, such as padding, it is desired that the length of both the input and target sequences that form the samples be consistent. Then, the input sequence length $L_{\mathrm{in}}$ is manually fixed and the output sequence length $L_{\mathrm{out}}$ is determined by the most representative target sequence length observed in the data, which in this case is two. The samples with $L_{\mathrm{out}}\neq 2$ are filtered out.Meaningful velocities. Given that the present research work focuses on road vehicles, all elements of input sequences with $v_{o}>400\;\frac{km}{h}$ are filtered out. This deletes outliers. Secondly, only samples with target sequences that contain velocities between $0\;\frac{km}{h}$ and $50\;\frac{km}{h}$ are considered. This covers the typical velocity range in urban traffic environments.Temporal consistence. As stated above, the OBD data cannot be obtained in deterministic time intervals. This implies atypical time lapses between consecutive measurements. In order to detect these atypical time lapses, each time lapse ${\Delta t}_{\mathrm{OBD}}$ between all consecutive OBD measurements contained in $T_{\mathrm{OBD}}$ is calculated. This results in $N_{\mathrm{OBD}}-1$ values. A mean $\mu_{\Delta t,\mathrm{OBD}}$ and a standard deviation $\sigma_{\Delta t,\mathrm{OBD}}$ are computed from all ${\Delta t}_{\mathrm{OBD}}$ values. Each ${\Delta t}_{\mathrm{OBD}}$ is compared to $\mu_{\Delta t,\mathrm{OBD}}$ and $\sigma_{\Delta t,\mathrm{OBD}}$. If $\mu_{\Delta t,\mathrm{OBD}}-\sigma_{\Delta t,\mathrm{OBD}}<{\Delta t}_{\mathrm{OBD}}<\mu_{\Delta t,\mathrm{OBD}}+\sigma_{\Delta t,\mathrm{OBD}}$, the sample where ${\Delta t}_{\mathrm{OBD}}$ belongs is kept and filtered out otherwise. The target sequence is filtered by analogy.

The dataset generated by the filtered samples is termed “primary sample set”.

#### 3.3.2. Data Balancing

This step aims to generate sample sets from the primary sample set generated in Section 3.3.1. This groups the samples according to predefined velocity ranges and equalize the number of samples of all groups. A graphic depiction of this is shown in Figure 8.

##### Sample Grouping by Velocity Range

All samples are grouped according to the mean velocity of the target sequence. Each group covers a range of $5\;\frac{km}{h}$, which results in a total of nine groups, where group 1 covers from $5\;\frac{km}{h}$ to $10\;\frac{km}{h}$ and group 9 covers from $45\;\frac{km}{h}$ to $50\;\frac{km}{h}$.

##### Data Shuffling

All samples within each group are shuffled so that samples from different driving tests are mixed with each other.

##### Data Trim

So as to have the same number of samples in all groups, the groups are trimmed to the size of the group with the least number of samples. This aids in avoiding training biases towards a specific group.

#### 3.3.3. Data Splitting

This step aims to generate the training $D_{\mathrm{train}}$, validation $D_{\mathrm{val}}$, and test $D_{\mathrm{test}}$ datasets that compose the ML-database $D_{\mathrm{ML}}$ required to train and evaluate the TNN. A graphic depiction of this is shown in Figure 9.

First, each of the groups generated in Section 3.3.2 is split so that 70% of its elements go into a training set, 15% of its elements to a validation set, and 15% to a test set. Then, all training sets are merged and shuffled so as to generate a single training dataset $D_{\mathrm{train}}$. The same is carried out for all validation sets in order to generate a single validation dataset $D_{\mathrm{val}}$ and for all test sets to obtain a single test dataset $D_{\mathrm{test}}$. The distribution of the training, validation, and test datasets end up with 46,964, 10,064, and 10,064 samples, respectively, for each one of the velocity groups.

A graphical depiction of the result of the data shuffling is shown in Figure 10.

### 3.4. Transformer Neural Network

In this paper, the used architecture of the TNN is based on the original Encoder–Decoder architecture proposed in [17] and shown in Figure 11. The task is a regression, where the network is trained to infer a sequence $\hat{\gamma }$ from an input sequence χ so that $\min\left(L\left(\hat{\gamma },\gamma \right)\right)$, where L expresses the loss function. In what follows, an overview of the model architecture and implementation details are given.

#### 3.4.1. Model Architecture

The network is trained in a recurrent manner. This means that in each iteration, a single element of the predicted sequence is inferred. This first inferred element is then added to the input sequence of the TNN-decoder to start the next prediction iteration. This is repeated until all $L_{\mathrm{out}}$ elements of the predicted sequence are obtained.

The input sequence is normalized to the [0, 1] range so that
(14)$$\begin{array}{c} v_{i,\mathrm{OBD}}^{\prime }=\frac{v_{i,\mathrm{OBD}}-\min\left({\tilde{\chi }}_{1}\right)}{\max\left({\tilde{\chi }}_{1}\right)-\min\left({\tilde{\chi }}_{1}\right)}, \end{array}$$
where ${\tilde{\chi }}_{1}$ contains the first elements of all input sequences of $D_{\mathrm{train}}$, $v_{i,\mathrm{OBD}}$ is a non-normalized value contained in ${\tilde{\chi }}_{1}$, and $v_{i,\mathrm{OBD}}^{\prime }$ is the normalized value of $v_{i,\mathrm{OBD}}$. This normalization process is repeated for the rest of the elements of the input sequence.

#### 3.4.2. Implementation Details

The main parameters selected in relation to the implemented TNN-based model are shown in Table 1.

##### Dataset and Batching

The dataset obtained after the sample generation (Section 3.2) and processing (Section 3.3) is partitioned into 70% for training, 15% for validation, and 15% for testing. The dataset in each of the three partitions is balanced in relation to the range of velocities contained in the samples. The number of samples of the dataset varies depending on $L_{\mathrm{in}}$. More details are given later (Section 4.4.1).

The selected batch size (number of samples per batch, $N_{\mathrm{samp}}$) varies from 16 to 64 for different experiments performed, so that in each experiment, the selected value is specified.

##### Loss Function

In this work, the Mean Squared Error is adopted as loss function L. Then, the computed loss for a batch $L_{\mathrm{batch}}$ is determined as follows
(15)$$\begin{array}{c} L_{\mathrm{batch}}=\frac{1}{N_{\mathrm{samp}}}\frac{1}{L_{\mathrm{out}}}\sum_{k=1}^{N_{\mathrm{samp}}}\sum_{t=1}^{L_{\mathrm{out}}}{\left(v_{t,k,\mathrm{REF}}-{\hat{v}}_{t,k,\mathrm{REF}}\right)}^{2}, \end{array}$$
where $v_{t,k,\mathrm{REF}}$ is the *t*-th element of the *k*-th target sequence and ${\hat{v}}_{t,k,\mathrm{REF}}$ is the *t*-th element of the *k*-th inferred sequence.

##### Regularization and Drop-Out

In the TNN model, each sub-layer of both the encoder and the decoder has a residual connection, which is completed by a normalization layer. Before these operations are performed, dropout is used as a regularization technique. A value of 0.1 is chosen following the original implementation during the training phase. The validation and testing of the model are performed with a dropout value equal to 0.

##### Optimizer

The model is trained via backpropagation with the Adam optimizer [33]. The learning rate lr is determined by the function $f_{lr}\left(\;\cdot \;\right)$ that starts with a warm-up phase and continues with a decaying learning rate afterwards. This strategy was previously investigated in [34,35] to assess its significance in training TNNs to help the network convergence. Constant values of the learning rate without a warm-up phase often causes the networks to quickly stagnate at local minima and not converge. The learning rate function $f_{lr}\left(\;\cdot \;\right)$ depends on the six parameters $lr_{\max}$, $lr_{\mathrm{start}}$, $n_{\mathrm{warmup}}$, $n_{\mathrm{drop}}$, drop, and $n_{\mathrm{epoch}}$ and is defined as shown in Equation (Equation 16).
(16)$$\begin{array}{c} lr=f_{lr}\left(\;\cdot \;\right)=\left\{\begin{array}{cc} \frac{lr_{\max}-lr_{\mathrm{start}}}{n_{\mathrm{warmup}}}\dot{(}n_{\mathrm{epoch}}-1), & n_{\mathrm{epoch}}\leq n_{\mathrm{warmup}}\\ \frac{lr_{\max}}{{\mathrm{drop}}^{\left(\frac{n_{\mathrm{epoch}}-1-n_{\mathrm{warmup}}}{n_{\mathrm{drop}}}\right)}}, & n_{\mathrm{epoch}}>n_{\mathrm{warmup}}. \end{array}\right. \end{array}$$

The selected parameters for the ADAM optimizer are $\beta_{1}=0.9$, $\beta_{2}=0.99$, and $\epsilon =10^{-9}$. The input parameters of the learning rate function $f_{lr}\left(\;\cdot \;\right)$ are indicated in the corresponding experiments.

##### Hardware

The model was trained on a computer running the Windows 10 operating system. It was equipped with an Intel Xeon W-2445 CPU processor with 3.7 GHz, 64 GB RAM, and a NVIDIA TITAN RTX GPU graphics card with 24 GB. The average run time for a single inference was 0.035 s during the evaluation. This time considers training with 422,680 (70%) samples, 90,574 samples (15%) for validation, and 90,574 samples (15%) for the test. The input data for each sample consider the OBD velocity as a feature. The input length is 5 for the baseline training and is varied for the input length ablation study. The learning target of each sample considers the reference sensor velocity as feature and has a length of 2.

## 4. Results

Once the ML-database was generated (Section 3.2) and prepared (Section 3.3), the next step was to evaluate the performance of the TNN to predict a sequence of future velocity data when it receives an input sequence composed by OBD measurements. In addition, ablation studies are conducted to obtain more details about the performance and robustness of the model against (1) the modification of the length of the input data and (2) the modification of the database size.

### 4.1. Evaluation Metrics

The performance of the ML-model is evaluated in terms of regression metrics. The following well-known evaluation metrics are considered: (1) Mean Absolute Error (MAE), (2) Root Mean Squared Error (RMSE), (3) $R^{2}$-Score ($R^{2}$), (4) Mean Percentage Error (MPE), and (5) Mean Absolute Percentage Error (MAPE). These metrics are defined in Equations (Equation 17)–(Equation 21) and consider the target value $v_{t,k,b,\mathrm{REF}}$, the mean of the targets values $\bar{v}$, and the predicted value ${\hat{v}}_{t,k,b,\mathrm{REF}}$:(17)$$\begin{array}{c} \mathrm{MAE}=\frac{1}{N_{\mathrm{batch}}}\frac{1}{N_{\mathrm{samp}}}\frac{1}{L_{\mathrm{out}}}\sum_{b=1}^{N_{\mathrm{batch}}}\sum_{k=1}^{N_{\mathrm{samp}}}\sum_{t=1}^{L_{\mathrm{out}}}\left|v_{t,k,b,\mathrm{REF}}-{\hat{v}}_{t,k,b,\mathrm{REF}}\right|, \end{array}$$
(18)$$\begin{array}{c} \mathrm{RMSE}=\sqrt{\frac{1}{N_{\mathrm{batch}}}\frac{1}{N_{\mathrm{samp}}}\frac{1}{L_{\mathrm{out}}}\sum_{b=1}^{N_{\mathrm{batch}}}\sum_{k=1}^{N_{\mathrm{samp}}}\sum_{t=1}^{L_{\mathrm{out}}}{\left(v_{t,k,b,\mathrm{REF}}-{\hat{v}}_{t,k,b,\mathrm{REF}}\right)}^{2}}, \end{array}$$
(19)$$R^{2}=1-\frac{\sum_{b=1}^{N_{\mathrm{batch}}}\sum_{k=1}^{N_{\mathrm{samp}}}\sum_{t=1}^{L_{\mathrm{out}}}{\left(v_{t,k,b,\mathrm{REF}}-{\hat{v}}_{t,k,b,\mathrm{REF}}\right)}^{2}}{\sum_{b=1}^{N_{\mathrm{batch}}}\sum_{k=1}^{N_{\mathrm{samp}}}\sum_{t=1}^{L_{\mathrm{out}}}{\left(v_{t,k,b,\mathrm{REF}}-\bar{v}\right)}^{2}},$$
(20)$$\begin{array}{c} \mathrm{MPE}=\;100\%\;\cdot \;\frac{1}{N_{\mathrm{batch}}}\frac{1}{N_{\mathrm{samp}}}\frac{1}{L_{\mathrm{out}}}\sum_{b=1}^{N_{\mathrm{batch}}}\sum_{k=1}^{N_{\mathrm{samp}}}\sum_{t=1}^{L_{\mathrm{out}}}\frac{v_{t,k,b,\mathrm{REF}}-{\hat{v}}_{t,k,b,\mathrm{REF}}}{v_{t,k,b,\mathrm{REF}}}, \end{array}$$
(21)$$\begin{array}{c} \mathrm{MAPE}=\;100\%\;\cdot \;\frac{1}{N_{\mathrm{batch}}}\frac{1}{N_{\mathrm{samp}}}\frac{1}{L_{\mathrm{out}}}\sum_{b=1}^{N_{\mathrm{batch}}}\sum_{k=1}^{N_{\mathrm{samp}}}\sum_{t=1}^{L_{\mathrm{out}}}\frac{|v_{t,k,b,\mathrm{REF}}-{\hat{v}}_{t,k,b,\mathrm{REF}}|}{v_{t,k,b,\mathrm{REF}}}, \end{array}$$
where $N_{\mathrm{batch}}$ indicates the number of batches per epoch.

### 4.2. Hyper-Parameter Optimization

The first experiments are performed with the objective of finding a combination of parameters for the learning rate function $f_{lr}\left(\;\cdot \;\right)$ that result in the lowest validation loss. Specifically, the parameters $lr_{\max}$ and drop are varied, such that $lr_{\max}\in \left\{10^{-3},10^{-4},10^{-5}\right\}$, drop∈{1.1,1.3,1.5}, and the remaining parameters are fixed at $lr_{\mathrm{start}}=10^{-12}$, $n_{\mathrm{warmup}}=5$ and $n_{\mathrm{drop}}=12.5$. Out of these ranges, nine parameter combinations are possible. The TNN is then trained with a database size of 603,828 samples, for 20 epochs, with a batch size of 64, an input sequence length of 5, and a target sequence length of 2. The evolution of the learning rate with respect to the training epochs can be seen in Figure 12. The training and validation losses for each epoch are shown in Figure 13 and Figure 14, respectively.

In view of the results shown in Figure 14, the parameters $lr_{\max}=10^{-4}$ and drop=1.1 show the lowest validation loss. Thus, these are the parameter values for the learning rate function for the rest of the experiments carried out in this work.

### 4.3. Baseline Training and Inference

Once the parameters for the learning rate function are determined, a quantitative evaluation of the TNN is performed. For this, the generalization and inference capability of the model for new samples is evaluated. This experiment considers the ML-database with 603,828 samples that are split into 70% training, 15% validation, and 15% test, 50 epochs, a batch size of 16, an input sequence length of 5, and a target sequence length of 2.

From the experiment described in the previous paragraph, 50 models were obtained: one per epoch. The model with the lowest validation loss during the validation step is considered as the best one; it is considered the baseline model for this work and is used for the test (inference) step. The evaluation metrics of the inference for the aforementioned model are shown in Table 2.

### 4.4. Ablation Studies

So as to determine the robustness of the model against variations in the input length and the number of training samples, different ablation studies were performed. In what follows, the quantitative results are shown.

#### 4.4.1. Modification of the Input Length

The first ablation study aimed to determine the performance of the model when the number of elements contained in the input sequence $L_{\mathrm{in}}$ was varied. The inference time required for the different cases was computed as well. The results are shown in Table 3, where the sequence length varies from 1 to 20, which results in a total of eight experiments.

It is important to note that as the length of the input sequence $L_{\mathrm{in}}$ increases, the number of training samples that can be generated from the ML-database decreases. This would imply a different number of training samples for each variation in $L_{\mathrm{in}}$. In order to perform all experiments with the same number of samples, and thus avoid $L_{\mathrm{in}}$-derived biases, the number of training samples is fixed for all experiments. Therefore, the total number of samples is set to 50,000, split into 70% training, 15% validation, and 15% test. All the experiments are performed with a maximum number of 20 training epochs, a batch size of 16, a target sequence length of two, and an input sequence length that varies for each experiment. The evaluation metrics are reported in Table 3.

#### 4.4.2. Modification of the Database Size

The ML-database generated and used in this paper consists of a large number of measurements. The objective of this ablation study is to determine the influence of the number of samples on the model performance. In this way, an adequate number of samples needed to obtain an acceptable model performance can be determined. This study consists of six experiments, where the ML-database varies from 1000 to 500,000, with an equal number of samples from each speed group defined in Section 3.3.2, and the data are split into 70% training, 15% validation, and 15% testing. Furthermore, all the experiments consider a maximum number of training epochs of 50, a batch size of 16, an input sequence length of five, and a target sequence length of two. Table 4 shows the results of the study.

### 4.5. Benchmark of the Transformer Neural Network

In order to contextualize the performance of the TNN to generate velocity correction data, two values are compared: (1) the MAE of the inference of the TNN, defined as MAE-TNN, and (2) the MAE when the OBD velocity data are compared to the reference sensor data, defined as MAE ($v_{o}$). To calculate MAE ($v_{o}$), the OBD velocity data $v_{o}$ are synchronized with the reference sensor data $v_{r}$ by using the UNIX time (see Section 3.2). Each OBD data point is then associated with the reference sensor measurement that is closest in time. Then, the velocity error $\eta_{\mathrm{OBD}}$ of a single OBD measurement is then defined as
(22)$$\begin{array}{c} \eta_{\mathrm{OBD}}=\left|v_{o}-v_{r}\right|. \end{array}$$

Thus, the MAE ($v_{o}$) is calculated as shown in Equation (Equation 23).
(23)$$\begin{array}{c} \mathrm{MAE}\;(v_{o})=\frac{1}{N}\sum_{i=1}^{N}\eta_{\mathrm{OBD}}, \end{array}$$
were *N* indicates the number of data points.

Then, the velocity errors per velocity group reported as MAE-TNN and MAE ($v_{o}$) are shown in Table 5, when considering the complete database. So as to gather more knowledge about the impact of the database size, the same comparison was performed, but when the TNN was trained with 1000 samples. This is shown in Table 6.

It should be noted that, as stated above, the OBD data cannot be obtained in deterministic time intervals, which leads to the fact that not every OBD measurement of the sensor dataset (Section 3.2) has a corresponding reference sensor data measurement. With the generated dataset, a total of 3,428,099 OBD measurements can be synchronized with a reference sensor measurement. The synchronization errors $\eta_{t}$ can also cause velocity errors. This maximum expected synchronization-derived velocity error $\eta_{v}$ is also computed. For this, a typical maximum acceleration for road vehicles of $9.8\;\frac{m}{s^{2}}$ is assumed. The synchronization errors derived from the used synchronization strategy and synchronization-derived velocity errors are shown in Table 7.

In addition, the EU regulation ([36]) defines the relation between the reference velocity $v_{r}$ and the OBD velocity $v_{o}$ as follows
(24)$$\begin{array}{c} 0\leq \left(v_{o}-v_{r}\right)\leq 0.1\;\cdot \;v_{r}+4\;\frac{km}{h}, \end{array}$$
which means that there is an absolute component ($+4\;\frac{km}{h}$), as well as a correction interval that can range from 0% to 10%. This is defined as such with the objective that the velocity shown to drivers is never less than the real velocity over ground with which the vehicle moves.

From the Equation (Equation 24), one can obtain an interval where the reference velocity could be with respect to the OBD velocity. An expert with knowledge about this regulation could then estimate an “expert velocity” $v_{e}$ from the OBD velocity under the assumption that this $v_{e}$ is most probably in the middle of said interval. The expert velocity would then be estimated as follows
(25)$$\begin{array}{c} v_{e}=\frac{v_{o}-2}{1.05}. \end{array}$$

The MAE for $v_{e}$ vs. $v_{r}$, defined as MAE ($v_{e}$), compared to the MAE of the inference of the TNN vs. $v_{r}$ are shown in Table 8 and Table 9. A comparison of $v_{e}$, $v_{r}$ and $v_{o}$ is shown in Figure 15.

In order to show the relation between the OBD and the reference sensor data, the synchronized OBD data are re-arranged by ordering the reference velocity in ascending order. Then, first, with respect to Equation (Equation 24), the relation $r_{v}$ between each reference velocity $v_{r}$ and its corresponding OBD measurement $v_{o}$ is defined, with respect to the upper limit of the interval indicated by the equation. This is obtained as follows:(26)$$\begin{array}{c} r_{v}=\frac{\frac{v_{o}-4}{1.1}}{v_{r}}. \end{array}$$

The results are shown in Figure 16 and Table 10.

Second, with respect to Equation (Equation 24), the relation $r_{v}^{*}$ between each reference velocity $v_{r}$ and its corresponding OBD measurement $v_{o}$ is defined, with respect to the lower limit of the interval indicated by Equation (Equation 27). The results are shown in Table 11 and Figure 17.
(27)$$\begin{array}{c} r_{v}^{*}=\frac{v_{o}}{v_{r}}. \end{array}$$

## 5. Discussion

The first aspect to highlight is that, as shown in Table 2, the MAE of the inference when using the complete generated database of 603,828 samples does not exceed $0.2\;\frac{km}{h}$, which accounts for $0.056\;\frac{m}{s}$. This value is practically identical to the accuracy of the Correvit S-Motion, which is one of the reference sensors. This implies that, with enough training data, the TNN is able to generate velocity correction data, as if they were delivered by the reference sensor itself.

Furthermore, it is interesting to note that, as shown in Table 4, the robustness of the method allows one to obtain highly accurate velocity correction data, even when smaller databases are used to train the TNN. The performed test with the worst performance is still able to deliver velocity correction data with an MAE of less than $0.9\;\frac{km}{h}$, which accounts for $0.25\;\frac{m}{s}$. This means that the accuracy of the velocity correction data that can be obtained from the TNN is barely different ($0.15\;\frac{m}{s}$ difference) than that of consumer-grade SatNav receivers [37]. This with a database size of 1000 samples.

Another proof of the robustness of the method can be inferred from Figure 15. There, the reference velocity $v_{r}$, the OBD velocity $v_{o}$, and expert velocity $v_{e}$ are shown. One can notice that $v_{o}$, and therefore $v_{e}$, present high noise levels. This can be explained by how the dataset is generated. As mentioned in Section 3.2, the test vehicle was driven randomly on the test track, including random and harsh acceleration and braking. This way of driving that fosters the variability of the dataset also causes larger differences between the reference and OBD velocity. However, the proposed method is able to learn the error patterns from the data and is still able to generate highly accurate velocity correction data from faulty OBD data.

One point that supports the use of ML-algorithms instead of classic mathematical methods is the expert velocity $v_{e}$. This velocity is calculated with two sources of information: the velocity range defined by the EU regulation and an OBD velocity sample. With these two pieces of information, one could argue (due to Gaussian distribution or statistics, for example) that the *real* velocity is probably in the middle of said range. Even though classic mathematical methods have the advantage of being interpretable, the proposed ML-method still outperforms what someone with expert knowledge about the applicable regulation and OBD access could deliver. This is shown in Table 8 and Table 9, and Figure 15. Here, one should remember that the sensor specifications, as well as how their measurements are processed and made available in the OBD, are unknown.

The next aspect to highlight is shown in Table 5. There, it can be seen that the MAE of the OBD data does increase as the velocity of the groups increases. This is consistent with the results shown in Table 11 and Figure 17, where there is a relation $r_{v}^{*}\approx 1.122$, which would make the OBD velocity $v_{o}$ diverge from the reference velocity $v_{r}$ as this last increases.

Supporting the results of Table 4, in Table 6 it can be seen that, even when one uses a 1000-sample database, the accuracy of the TNN to generate velocity-correction data vastly outperforms the use of the raw OBD data. This is proven as the MAE of the TNN for all velocity groups is ≈74% better than the raw OBD data: $0.862\;\frac{km}{h}$ for the TNN versus $3.480\;\frac{km}{h}$ for the OBD.

An important aspect that can be derived from Table 8 and Table 9 is that, even in a worst-case scenario where the TNN is trained using a small database of 1000 samples, the mean absolute error of the TNN-inferred velocity is smaller (6.9%) than what an expert with knowledge about the EU regulation could generate. On average, the benefit of using a TNN becomes more evident as the database size is increased. When a database of 603,828 samples is used, the TNN-inferred velocity is much better (81.98% improvement) than the velocity that an expert could generate.

From Table 10 and Table 11 and Figure 16 and Figure 17, it can be inferred that, regardless of whether one considers $r_{v}^{*}$ or $r_{v}$, there is a somewhat constant relation between $v_{o}$ and $v_{r}$. Nevertheless, this is not a mathematical relation that can be expressed by a formula as it is neither a constant offset nor a fixed percentage, but, as shown in Table 10 and Table 11, this relation presents large fluctuations. This further supports the use of ML-methods, as the TNN is able to learn a proper relation between $v_{o}$ and $v_{o}$, including whichever sensor noise is present in the training data.

Another aspect worth discussing is that, as shown in Table 3, the input length does not really provoke a decay in the accuracy of the inference of the TNN. Future work could include an analysis to determine why the smallest length size of 1 appears to have the lowest MAE of all the tests.

In the different experiments performed, the runtime is measured, i.e., the time taken by the ML-model to make a single inference with the best performing model. Thus, a runtime of ≈40 ms is observed for the baseline training with the mentioned hardware. This suggests that future work could include the real-time implementation of the proposed method based on TNN.

As for the monetary aspect, the present research work serves as proof of concept, where the proposed method is analyzed in depth for its advantages and limitations. Such an analysis requires high-end reference sensors and computing units. As understood from the results, the performance of the method to deliver accurate velocity correction data closely resembles that of the reference sensors themselves. Future work would then be to analyze the downscaling effects, that is, when other less accurate sensors are used as learn target, or when the training and inference of the TNN are performed in resource-limited platforms. Furthermore, the authors agree that especially for the data acquisition and training phase, especially with very large datasets, a dedicated hardware solution is required and would be expensive. However, the inference phase for velocity estimation given new measurements using a trained model can be carried out with traditional processing units such as Central Processing Units, with the help of other architectures such as Field-Programmable Gate Arrays, instead of using Graphics Processing Units, as the latter solutions are generally more expensive [38].

Finally, from the present research work, one limitation of the proposed method that could be identified is the update rate of the OBD data. As explained in Section 3.2, the OBD data cannot be obtained in deterministic intervals as a consequence of the CAN protocol. The OBD velocity will be overwritten by any other more relevant information. If no velocity can be obtained from the OBD, no velocity correction data can be generated.

## 6. Conclusions

This research work makes the contributions of (1) proposing an ML-model to generate highly accurate velocity correction data from real-world OBD data and (2) performing various ablation studies that confirm the robustness of the method against database size and input sequence length variations.

The ML-model, a TNN in this case, is used for the generation of correction data for INSs from OBD data obtained from commercial vehicles. This is with the objective of estimating a highly accurate vehicle state that can be used for the assessment and validation of AD and ADAS. The focus was set on velocity-correction data.

The method applied in the study covers all stages of a typical regression problem within the supervised learning paradigm. It starts with data generation, by recording data with a test vehicle on the CARISSMA tracks (Germany), which covers ≈535 km for a duration of ≈1282 min. Subsequently, the generated data is processed (data selection, data balancing, and data splitting) so that an in-house ML-database is generated. Finally, the in-house database is used to train an ML-model based on TNN, and the inferences obtained from the TNN are evaluated against state-of-the-art reference sensors.

The ML-model is able to learn the mapping from the OBD data to the reference sensor measurements. The test results demonstrate that the method is able to make predictions with the velocity over ground of the vehicle with an average MAE of ≈$0.167\;\frac{km}{h}$. Therefore, the results of this investigation highlight the potential usefulness of TNN, originally designed for NLP tasks, for the generation of velocity correction data for INSs from OBD data. The results also show that the proposed method outperforms the velocity that an expert could generate with the available data. Additionally, the inference time for the generation of new correction data is always below 40 ms, suggesting the capability of a real-time implementation.

This study covers the range of typical speeds in urban traffic environments, up to $50\;\frac{km}{h}$. This is due to driving limitations on the test track, presenting difficulties acquiring enough training data at velocities higher than $50\;\frac{km}{h}$. However, the method is not limited to these velocities, but applies to higher velocities as well. Future work could include gathering data at highway velocities to corroborate the accuracy of the TNN to generate velocity-correction data. Other aspects that could be analyzed in future work are the consideration of different weather conditions, adaptive learning online, or varying road conditions.

It should be noted that diversity in applications to multiple vehicles is not included in this work because the main objective is to present the methodology starting from the data acquisition to the generation of correction data in the ML inference phase. The robustness of the method using the test vehicle is supported by the results provided. However, the method is not exclusive to the vehicle used but can be extended to different types of vehicles. In this regard, future research is conceived to evaluate the use of transfer learning so that a pre-trained model is used as a general solution and fine-tuning is applied to adapt the model to each specific vehicle.

Finally, only the velocity over the ground was used to generate velocity-correction data. Future work could also include the analysis of other sensors (such as the wheel-speed sensors or the steering-wheel angle) to generate correction data for other state variables.

## Figures and Tables

**Figure 1 sensors-23-00159-f001:**
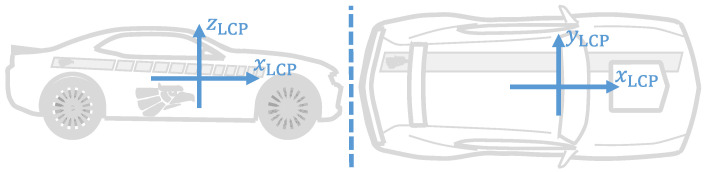
Graphical depiction of the Local Car Plane.

**Figure 2 sensors-23-00159-f002:**
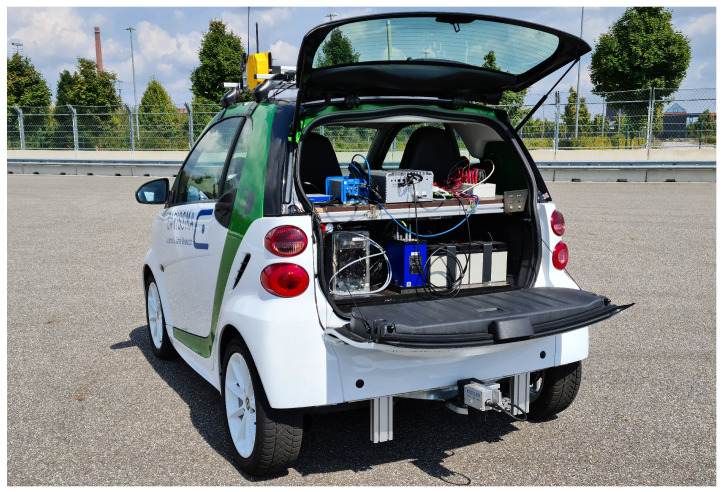
Test vehicle equipped with (1) an ADMA in the trunk (blue), (2) a Correvit sensor on the trailer hitch (gray), (3) a car PC in the trunk (gray), (4) a SatNav receiver on the roof (gray), and (5) a radio antenna for RTK correction data on the roof (black).

**Figure 3 sensors-23-00159-f003:**
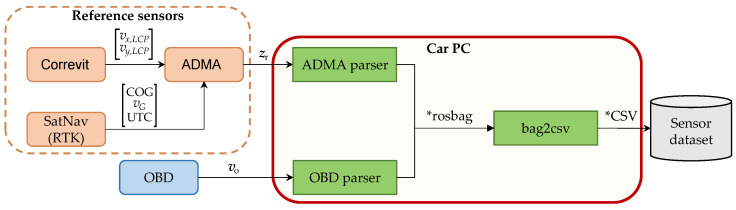
Methodology for data generation.

**Figure 4 sensors-23-00159-f004:**
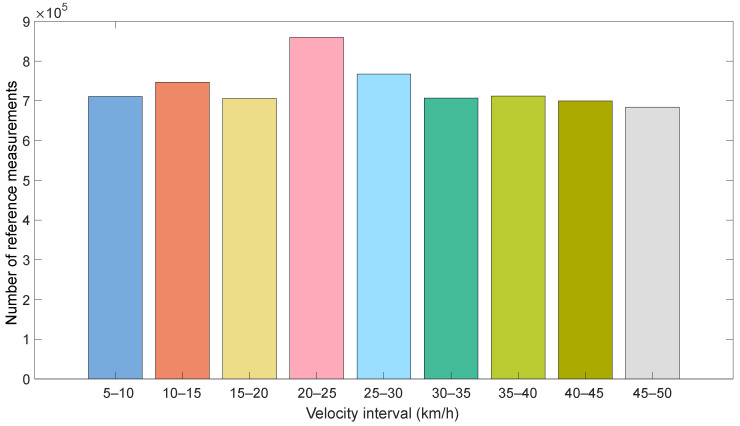
Histogram of the generated dataset. Each color indicates a velocity range of the reference data.

**Figure 5 sensors-23-00159-f005:**
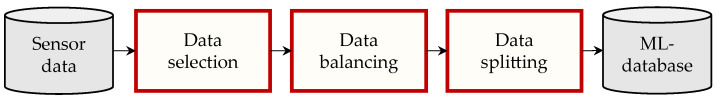
Methodology for the data processing.

**Figure 6 sensors-23-00159-f006:**
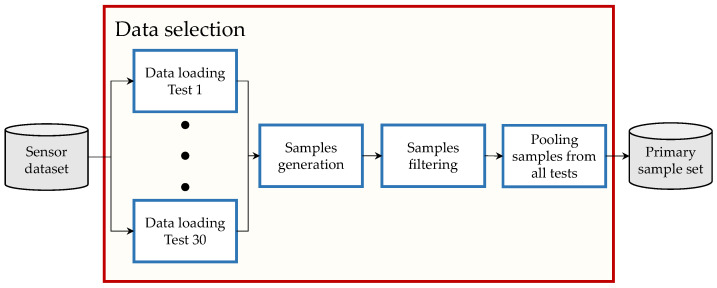
Overview of the data selection process.

**Figure 7 sensors-23-00159-f007:**
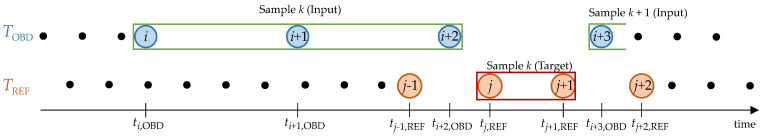
Graphical depiction of the sample generation for $L_{\mathrm{in}}=3$.

**Figure 8 sensors-23-00159-f008:**
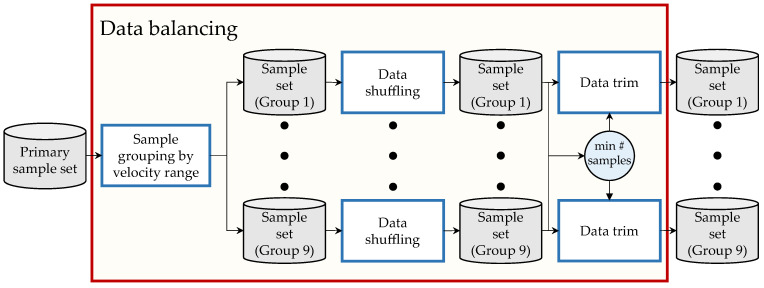
Overview of the data balancing process.

**Figure 9 sensors-23-00159-f009:**
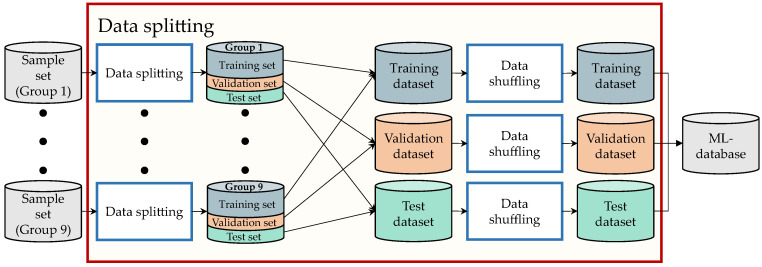
Overview of the data splitting process.

**Figure 10 sensors-23-00159-f010:**
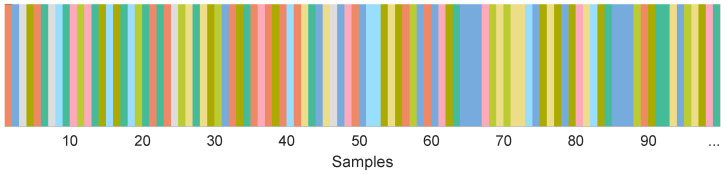
The order of the first 100 training samples. Each color indicates the target velocity range as shown in Figure 4.

**Figure 11 sensors-23-00159-f011:**
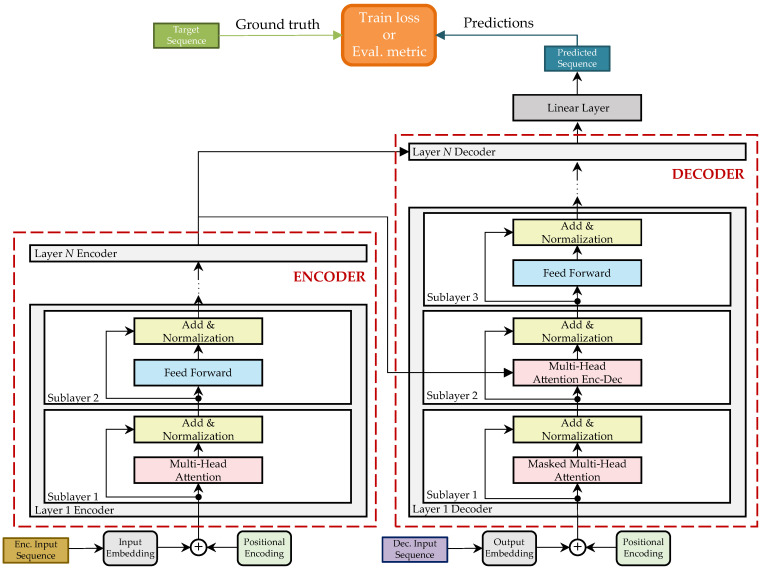
Model architecture of the TNN based on [17].

**Figure 12 sensors-23-00159-f012:**
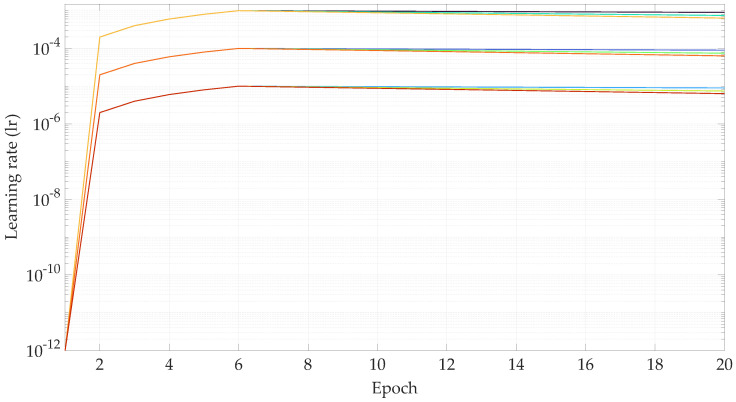
Learning rate with respect of the training epoch for the hyper-parameter optimization. $\left[lr_{\max}=10^{-3},\mathrm{drop}=1.1\right]$, $\left[lr_{\max}=10^{-4},\mathrm{drop}=1.1\right]$, $\left[lr_{\max}=10^{-5},\mathrm{drop}=1.1\right]$, $\left[lr_{\max}=10^{-3},\mathrm{drop}=1.3\right]$, $\left[lr_{\max}=10^{-4},\mathrm{drop}=1.3\right]$, $\left[lr_{\max}=10^{-5},\mathrm{drop}=1.3\right]$, $\left[lr_{\max}=10^{-3},\mathrm{drop}=1.5\right]$, $\left[lr_{\max}=10^{-4},\mathrm{drop}=1.5\right]$, and $\left[lr_{\max}=10^{-5},\mathrm{drop}=1.5\right]$.

**Figure 13 sensors-23-00159-f013:**
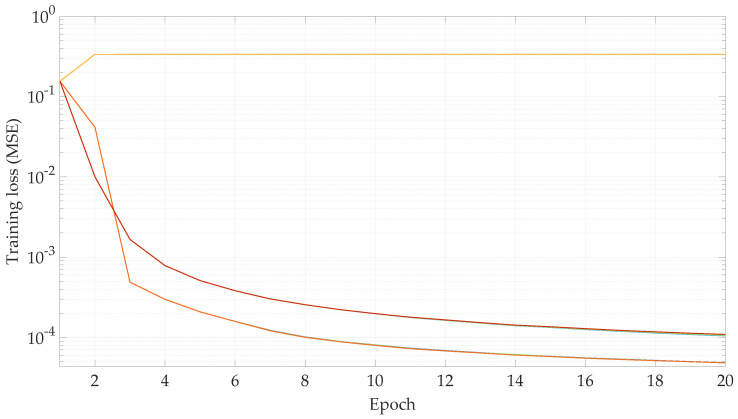
Training loss with respect of the training epoch for the hyper-parameter optimization. $\left[lr_{\max}=10^{-3},\mathrm{drop}=1.1\right]$, $\left[lr_{\max}=10^{-4},\mathrm{drop}=1.1\right]$, $\left[lr_{\max}=10^{-5},\mathrm{drop}=1.1\right]$, $\left[lr_{\max}=10^{-3},\mathrm{drop}=1.3\right]$, $\left[lr_{\max}=10^{-4},\mathrm{drop}=1.3\right]$, $\left[lr_{\max}=10^{-5},\mathrm{drop}=1.3\right]$, $\left[lr_{\max}=10^{-3},\mathrm{drop}=1.5\right]$, $\left[lr_{\max}=10^{-4},\mathrm{drop}=1.5\right]$, and $\left[lr_{\max}=10^{-5},\mathrm{drop}=1.5\right]$.

**Figure 14 sensors-23-00159-f014:**
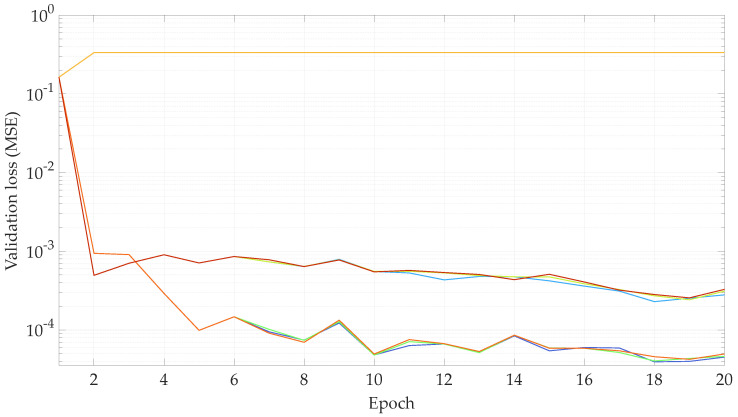
Validationloss with respect of the validation epoch for the hyper-parameter optimization. $\left[lr_{\max}=10^{-3},\mathrm{drop}=1.1\right]$, $\left[lr_{\max}=10^{-4},\mathrm{drop}=1.1\right]$, $\left[lr_{\max}=10^{-5},\mathrm{drop}=1.1\right]$, $\left[lr_{\max}=10^{-3},\mathrm{drop}=1.3\right]$, $\left[lr_{\max}=10^{-4},\mathrm{drop}=1.3\right]$, $\left[lr_{\max}=10^{-5},\mathrm{drop}=1.3\right]$, $\left[lr_{\max}=10^{-3},\mathrm{drop}=1.5\right]$, $\left[lr_{\max}=10^{-4},\mathrm{drop}=1.5\right]$, and $\left[lr_{\max}=10^{-5},\mathrm{drop}=1.5\right]$.

**Figure 15 sensors-23-00159-f015:**
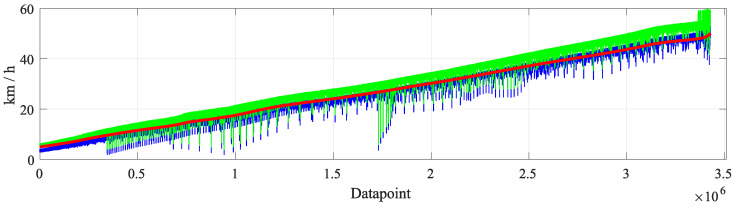
Comparisonbetween the reference $\left(v_{r}\right)$, expert$\left(v_{e}\right)$ and OBD$\left(v_{o}\right)$ velocities.

**Figure 16 sensors-23-00159-f016:**
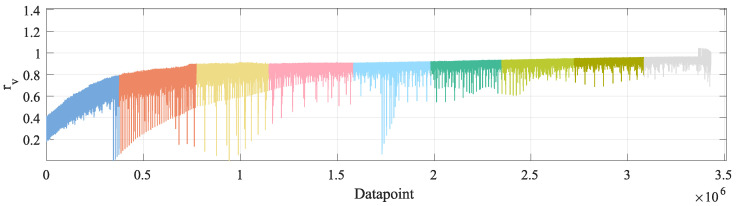
Mathematical relation $r_{v}$ for the different velocity groups: $5\;\frac{km}{h}$–$10\;\frac{km}{h}$, $10\;\frac{km}{h}$–$15\;\frac{km}{h}$, $15\;\frac{km}{h}$–$20\;\frac{km}{h}$, $20\;\frac{km}{h}$–$25\;\frac{km}{h}$, $25\;\frac{km}{h}$–$30\;\frac{km}{h}$, $30\;\frac{km}{h}$–$35\;\frac{km}{h}$, $35\;\frac{km}{h}$–$40\;\frac{km}{h}$, $40\;\frac{km}{h}$–$45\;\frac{km}{h}$, $45\;\frac{km}{h}$–$50\;\frac{km}{h}$.

**Figure 17 sensors-23-00159-f017:**
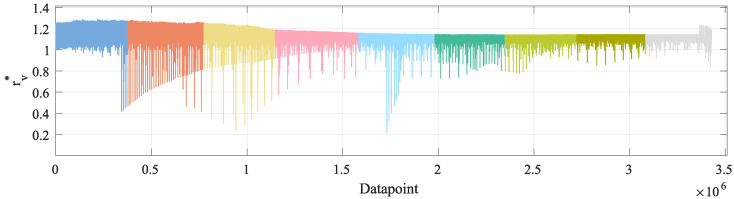
Mathematical relation $\frac{v_{o}}{v_{r}}$ for the different velocity groups: $5\;\frac{km}{h}$–$10\;\frac{km}{h}$, $10\;\frac{km}{h}$–$15\;\frac{km}{h}$, $15\;\frac{km}{h}$–$20\;\frac{km}{h}$, $20\;\frac{km}{h}$–$25\;\frac{km}{h}$, $25\;\frac{km}{h}$–$30\;\frac{km}{h}$, $30\;\frac{km}{h}$–$35\;\frac{km}{h}$, $35\;\frac{km}{h}$–$40\;\frac{km}{h}$, $40\;\frac{km}{h}$–$45\;\frac{km}{h}$, $45\;\frac{km}{h}$–$50\;\frac{km}{h}$.

**Table 1 sensors-23-00159-t001:** Selected parameters of the model based on TNN.

Parameter	Value
# Encoder layers	6
# Encoder sub-layers	2
# Decoder layers	6
# Decoder sub-layers	3
# Attention heads	8
# Hidden dimension Feed Forward Network	2048
# Model dimension	512
# Attention queries dimension	64
# Attention keys dimension	64
# Attention values dimension	64

**Table 2 sensors-23-00159-t002:** Evaluation metrics for the baseline model for each of the velocity ranges.

Group	MAE $\left(\frac{km}{h}\right)$	RMSE $\left(\frac{km}{h}\right)$	$R^{2}$	MPE $\left(\%\right)$	MAPE $\left(\%\right)$
$5\;\frac{km}{h}$–$10\;\frac{km}{h}$	0.129	0.191	0.984	−0.173	0.020
$10\;\frac{km}{h}$–$15\;\frac{km}{h}$	0.122	0.221	0.973	−0.004	0.010
$15\;\frac{km}{h}$–$20\;\frac{km}{h}$	0.145	0.217	0.977	−0.001	0.080
$20\;\frac{km}{h}$–$25\;\frac{km}{h}$	0.192	0.252	0.967	0.003	0.008
$25\;\frac{km}{h}$–$30\;\frac{km}{h}$	0.180	0.237	0.974	0.005	0.007
$30\;\frac{km}{h}$–$35\;\frac{km}{h}$	0.181	0.251	0.970	0.004	0.006
$35\;\frac{km}{h}$–$40\;\frac{km}{h}$	0.176	0.247	0.971	0.003	0.005
$40\;\frac{km}{h}$–$45\;\frac{km}{h}$	0.197	0.269	0.964	0.003	0.005
$45\;\frac{km}{h}$–$50\;\frac{km}{h}$	0.181	0.262	0.941	0.003	0.004
$5\;\frac{km}{h}$–$50\;\frac{km}{h}$	0.167	0.240	0.999	0.001	0.008

**Table 3 sensors-23-00159-t003:** Evaluation metrics for the TNN model when trained with input sequences of different lengths.

Input Length	MAE	RMSE	$R^{2}$	MPE	MAPE	Time
$L_{\mathrm{in}}$	$\left(\frac{km}{h}\right)$	$\left(\frac{km}{h}\right)$		$\left(\%\right)$	$\left(\%\right)$	$\mu \pm \sigma \;\left(s\right)$
1	0.278	0.361	0.999	−0.001	0.015	0.0325 ± 0.0037
2	0.472	0.604	0.998	0.015	0.023	0.0341 ± 0.0036
3	0.438	0.545	0.998	0.017	0.023	0.0340 ± 0.0036
4	0.333	0.419	0.999	0.002	0.019	0.0345 ± 0.0035
5	0.327	0.435	0.999	−0.012	0.017	0.0349 ± 0.0038
10	0.360	0.453	0.999	−0.008	0.018	0.0356 ± 0.0030
15	0.362	0.458	0.999	0.010	0.017	0.0356 ± 0.0027
20	0.374	0.481	0.999	−0.007	0.016	0.0377 ± 0.0032

**Table 4 sensors-23-00159-t004:** Evaluation metrics for the TNN model when trained with different database sizes.

Database Size	MAE $\left(\frac{km}{h}\right)$	RMSE $\left(\frac{km}{h}\right)$	$R^{2}$	MPE $\left(\%\right)$	MAPE $\left(\%\right)$
1000	0.863	1.037	0.999	−0.017	0.057
5000	0.482	0.570	0.998	−0.002	0.028
10,000	0.252	0.310	0.998	−0.007	0.011
50,000	0.251	0.312	0.999	0.005	0.013
100,000	0.223	0.320	0.999	−0.002	0.010
500,000	0.210	0.273	0.999	−0.002	0.010
603,828	0.167	0.240	0.999	−0.001	0.008

**Table 5 sensors-23-00159-t005:** MAE for the OBD vs. reference sensor comparison and the MAE for the inference of the TNN vs. reference sensor comparison. Database size: 603,828.

Group	MAE-TNN $\left(\frac{km}{h}\right)$	MAE($v_{o}$) $\left(\frac{km}{h}\right)$	Error Variation %
$5\;\frac{km}{h}$–$10\;\frac{km}{h}$	0.129	0.753	−82.87
$10\;\frac{km}{h}$–$15\;\frac{km}{h}$	0.122	1.462	−91.48
$15\;\frac{km}{h}$–$20\;\frac{km}{h}$	0.145	2.135	−93.21
$20\;\frac{km}{h}$–$25\;\frac{km}{h}$	0.192	2.812	−93.17
$25\;\frac{km}{h}$–$30\;\frac{km}{h}$	0.180	3.385	−94.68
$30\;\frac{km}{h}$–$35\;\frac{km}{h}$	0.181	4.052	−95.53
$35\;\frac{km}{h}$–$40\;\frac{km}{h}$	0.176	4.735	−96.28
$40\;\frac{km}{h}$–$45\;\frac{km}{h}$	0.197	5.434	−96.37
$45\;\frac{km}{h}$–$50\;\frac{km}{h}$	0.181	6.248	−97.10
$5\;\frac{km}{h}$–$50\;\frac{km}{h}$	0.167	3.480	−95.07

**Table 6 sensors-23-00159-t006:** MAE for the OBD vs. reference sensor comparison and the MAE for the inference of the TNN vs. reference sensor comparison. Database size: 1000.

Group	MAE-TNN $\left(\frac{km}{h}\right)$	MAE($v_{o}$) $\left(\frac{km}{h}\right)$	Error Variation %
$5\;\frac{km}{h}$–$10\;\frac{km}{h}$	1.864	0.753	+147.54
$10\;\frac{km}{h}$–$15\;\frac{km}{h}$	0.360	1.462	−74.86
$15\;\frac{km}{h}$–$20\;\frac{km}{h}$	0.630	2.135	−70.49
$20\;\frac{km}{h}$–$25\;\frac{km}{h}$	0.744	2.812	−73.54
$25\;\frac{km}{h}$–$30\;\frac{km}{h}$	0.992	3.385	−70.69
$30\;\frac{km}{h}$–$35\;\frac{km}{h}$	1.040	4.052	−74.33
$35\;\frac{km}{h}$–$40\;\frac{km}{h}$	0.588	4.735	−87.58
$40\;\frac{km}{h}$–$45\;\frac{km}{h}$	0.946	5.434	−82.59
$45\;\frac{km}{h}$–$50\;\frac{km}{h}$	0.603	6.248	−90.35
$5\;\frac{km}{h}$–$50\;\frac{km}{h}$	0.863	3.480	−74.51

**Table 7 sensors-23-00159-t007:** Synchronization errors between an OBD and a reference sensor measurement. # of measurements: 3,428,099.

Group	$\min\left(\eta_{t}\right)$	mean $\left(\eta_{t}\right)$	$\max\left(\eta_{t}\right)$	$\max\left(\eta_{v}\right)$
$\left(\frac{km}{h}\right)$	$\left(s\right)$	$\left(s\right)$	$\left(s\right)$	$\left(\frac{km}{h}\right)$
$5\;\frac{km}{h}$–$10\;\frac{km}{h}$	0.0	0.0025	0.0098	0.346
$10\;\frac{km}{h}$–$15\;\frac{km}{h}$	0.0	0.0024	0.0099	0.349
$15\;\frac{km}{h}$–$20\;\frac{km}{h}$	0.0	0.0025	0.0099	0.349
$20\;\frac{km}{h}$–$25\;\frac{km}{h}$	0.0	0.0025	0.0099	0.349
$25\;\frac{km}{h}$–$30\;\frac{km}{h}$	0.0	0.0025	0.0097	0.342
$30\;\frac{km}{h}$–$35\;\frac{km}{h}$	0.0	0.0025	0.0098	0.346
$35\;\frac{km}{h}$–$40\;\frac{km}{h}$	0.0	0.0025	0.0099	0.349
$40\;\frac{km}{h}$–$45\;\frac{km}{h}$	0.0	0.0025	0.0092	0.325
$45\;\frac{km}{h}$–$50\;\frac{km}{h}$	0.0	0.0025	0.0099	0.349
$5\;\frac{km}{h}$–$50\;\frac{km}{h}$	0.0	0.0025	0.0099	0.349

**Table 8 sensors-23-00159-t008:** MAE for $v_{e}$ vs. $v_{r}$ and the MAE for the inference of the TNN vs. $v_{r}$. Database size: 603,828.

Group	MAE-TNN $\left(\frac{km}{h}\right)$	MAE ($v_{e}$) $\left(\frac{km}{h}\right)$	Error Variation %
$5\;\frac{km}{h}$–$10\;\frac{km}{h}$	0.129	1.539	−91.62
$10\;\frac{km}{h}$–$15\;\frac{km}{h}$	0.122	1.108	−88.99
$15\;\frac{km}{h}$–$20\;\frac{km}{h}$	0.145	0.706	−79.46
$20\;\frac{km}{h}$–$25\;\frac{km}{h}$	0.192	0.365	−47.40
$25\;\frac{km}{h}$–$30\;\frac{km}{h}$	0.180	0.180	0.00
$30\;\frac{km}{h}$–$35\;\frac{km}{h}$	0.181	0.436	−58.49
$35\;\frac{km}{h}$–$40\;\frac{km}{h}$	0.176	0.823	−78.61
$40\;\frac{km}{h}$–$45\;\frac{km}{h}$	0.197	1.256	−84.62
$45\;\frac{km}{h}$–$50\;\frac{km}{h}$	0.181	1.809	−89.99
$5\;\frac{km}{h}$–$50\;\frac{km}{h}$	0.167	0.927	−81.98

**Table 9 sensors-23-00159-t009:** MAE for $v_{e}$ vs. $v_{r}$ and the MAE for the inference of the TNN vs. $v_{r}$. Database size: 1000.

Group	MAE-TNN $\left(\frac{km}{h}\right)$	MAE ($v_{e}$) $\left(\frac{km}{h}\right)$	Error Variation %
$5\;\frac{km}{h}$–$10\;\frac{km}{h}$	1.864	1.539	21.12
$10\;\frac{km}{h}$–$15\;\frac{km}{h}$	0.360	1.108	−67.51
$15\;\frac{km}{h}$–$20\;\frac{km}{h}$	0.630	0.706	−10.76
$20\;\frac{km}{h}$–$25\;\frac{km}{h}$	0.744	0.365	103.84
$25\;\frac{km}{h}$–$30\;\frac{km}{h}$	0.992	0.180	451.11
$30\;\frac{km}{h}$–$35\;\frac{km}{h}$	1.040	0.436	138.53
$35\;\frac{km}{h}$–$40\;\frac{km}{h}$	0.588	0.823	−28.55
$40\;\frac{km}{h}$–$45\;\frac{km}{h}$	0.946	1.256	−24.68
$45\;\frac{km}{h}$–$50\;\frac{km}{h}$	0.603	1.809	−66.67
$5\;\frac{km}{h}$–$50\;\frac{km}{h}$	0.863	0.927	−6.90

**Table 10 sensors-23-00159-t010:** Relation $r_{v}$ according to Equation (Equation 26). # of measurements: 3,428,099.

Group $\left(\frac{km}{h}\right)$	$\min\left(r_{v}\right)$	mean $\left(r_{v}\right)$	$\max\left(r_{v}\right)$	$\Delta \left(r_{v}*\right)$ $\left(\max\left(r_{v}\right)-\min\left(r_{v}\right)\right)$
$5\;\frac{km}{h}$–$10\;\frac{km}{h}$	0.001	0.486	0.790	0.789
$10\;\frac{km}{h}$–$15\;\frac{km}{h}$	0.070	0.718	0.898	0.828
$15\;\frac{km}{h}$–$20\;\frac{km}{h}$	0.001	0.809	0.912	0.911
$20\;\frac{km}{h}$–$25\;\frac{km}{h}$	0.347	0.861	0.911	0.564
$25\;\frac{km}{h}$–$30\;\frac{km}{h}$	0.062	0.889	0.921	0.858
$30\;\frac{km}{h}$–$35\;\frac{km}{h}$	0.546	0.910	0.934	0.388
$35\;\frac{km}{h}$–$40\;\frac{km}{h}$	0.608	0.927	0.949	0.341
$40\;\frac{km}{h}$–$45\;\frac{km}{h}$	0.689	0.939	0.959	0.271
$45\;\frac{km}{h}$–$50\;\frac{km}{h}$	0.689	0.952	1.043	0.353
$5\;\frac{km}{h}$–$50\;\frac{km}{h}$	0.001	0.833	1.043	1.042

**Table 11 sensors-23-00159-t011:** Relation $r_{v}^{*}$ according to Equation (Equation 27). # of measurements: 3,428,099.

Group $\left(\frac{km}{h}\right)$	$\min\left(r_{v}\right)$	mean $\left(r_{v}\right)$	$\max\left(r_{v}\right)$	$\Delta \left(r_{v}*\right)$ $\left(\max\left(r_{v}\right)-\min\left(r_{v}\right)\right)$
$5\;\frac{km}{h}$–$10\;\frac{km}{h}$	0.0	1.1001	1.2855	1.2855
$10\;\frac{km}{h}$–$15\;\frac{km}{h}$	0.0	1.1173	1.2805	1.2805
$15\;\frac{km}{h}$–$20\;\frac{km}{h}$	0.0124	1.1240	1.2518	1.2394
$20\;\frac{km}{h}$–$25\;\frac{km}{h}$	0.0168	1.1242	1.1911	1.1743
$25\;\frac{km}{h}$–$30\;\frac{km}{h}$	0.0280	1.1233	1.1572	1.1291
$30\;\frac{km}{h}$–$35\;\frac{km}{h}$	0.7319	1.1249	1.1469	0.4149
$35\;\frac{km}{h}$–$40\;\frac{km}{h}$	0.7800	1.1260	1.1454	0.3654
$40\;\frac{km}{h}$–$45\;\frac{km}{h}$	0.8544	1.1282	1.1464	0.2920
$45\;\frac{km}{h}$–$50\;\frac{km}{h}$	0.8390	1.1329	1.2304	0.3914
$5\;\frac{km}{h}$–$50\;\frac{km}{h}$	0.0	1.1222	1.2855	1.2855

## Data Availability

Not applicable.

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
