# Peer review of "Generation of Correction Data for Autonomous Driving by Means of Machine Learning and On-Board Diagnostics"

_sensors, 2022, doi:10.3390/s23010159_

Round 1

Reviewer 1 Report

This paper propose a ML-model to generate highly accurate velocity correction data from real-world OBD data. The work contains some interesting innovative point, it can be seen that the article has done very substantial research work. Overall, the article is well organized, and its presentation is good, which can be accepted in present form. 

Reviewer 2 Report

This paper proposed a methodology that consisted on the use of a machine learning  method with the objective of generating highly accurate velocity correction data from on-board diagnostics data. It is an interesting topic. The paper was well written and is easy to follow. It may be improved from the following points, mainly in the Discussion:

1. Please kindly provide more discussions on the practical implications of the study

2. Please give some discussions on the possible limitations of the study

Reviewer 3 Report

The manuscript is dealing with the challenging problem of obtaining accurate information on the vehicle states in challenging environments - when GNSS-based information can be of lower quality or not available.

The manuscript is based on a large dataset and contains a significant amount of results that should be of interest to a wide range of readers.

A few general comments first for which the reviewer would like to have a discussions added to the manuscript:

- the manuscript is fairly large and it is not always clear if the level of details provided is justified. For example, there are parts in the related works section (e.g. p.4) that are deviating from the main topic and discussing rather generic ML applications for the ground vehicles; the data exploration and pre-processing part is presented on ~6 pages and there is a feeling it can be more condensed without affecting the quality, readability and transparency of the manuscript.

- from the other side, the discussion of the ground truth information could be further enhanced. Was the quality of the GT evaluated? It appears there is no such discussion and the only reference-related information available is on Correvit S-Motion accuracy and while the link in the references section does not work it is difficult to understand if the system is able to maintain performance in all test conditions.  It should be noted also that the reference system presented in Figure 3 has another component with unspecified contribution to the overall reference system performance. In such conditions it is difficult to evaluate margins of the true performance of the proposed solution.

- the dataset collected appears to be large, however, there is little mention of the diversity of the data. For example, if there is insufficient diversity in terms of trajectories and vehicle motion available in the data one would expect ML-based system to produce overly optimistic results with unknow ability to generalize.

The reviewer would like also to ask for clarifications on several aspects in the presented manuscript:

- the performance metrics used require some discussion. E.g. why both MSE and RMSE selected? What advantages one would get from MPE and MAPE analysis? From another perspective a commonly used R2 regression performance metric is not mentioned

- It appears there are some issues with the data in the tables provided in Section 4. In particular, MAE and RMSE are having exactly same values which is rather unusual, MSE is presented with incorrect units (should be squared), MPE and MAPE are expected to be in percents as per Eqs 20 and 21.

- a discussion authors have on the expert knowledge is interesting and insightful, however, Eq 23 which is used to derive the consequent estimates is providing an interval for corrections rather than exact value. It would be good to add the corresponding discussion in the manuscript and refine the results presented as appropriate.

-the manuscript discusses cost of the enhanced sensors as one of the arguments towards using ML-based algorithms instead. It would be good to have also discussion of the added cost from the computing solution. The one used in the present study would be fairly expensive and the cheaper ones may not provide necessary computing power.

-the results in figures 18 - 20 demonstrate some sharp falls. What is the nature of these falls? It would be also good to see more clearly if they are present in all dependencies (e.g. in Figure 18)

Finally, while the language used is at the good technical level (except for a minor cases where word "derive" is used in rather unusual way) the use of Shown is in captions for all Tables and Figures seems to be redundant. In reviewers opinion it can be safely removed and the remaining parts of captions will still be clear and representative.

Round 2

Reviewer 3 Report

The comments have been adequately addressed. Many thanks to the authors for their efforts.